# NeuroSlice: Forward Selection-Based LLM Pruning via Neuron Contribution Decomposition

## Abstract

Large language models (LLMs) have dramatically advanced natural language processing, but their deployment is often hindered by exorbitant computational and memory demands. LLM pruning offers a promising pathway to efficiency, yet most pruning methods rely on the layer output as the signal for parameter importance estimation. In this work, we revisit this issue and demonstrate that the layer output is not an atomic unit. Leveraging matrix identity transformations, we decompose each layer's output into an additive summation of individual neuron contributions, thereby reshaping the original token-by-feature tensor into a more granular token-by-feature-by-neuron representation. This decomposition yields much richer pruning signals by explicitly quantifying each neuron's individual contribution to the original layer output, enabling us to convert structured pruning as an neuron subset selection problem. To further optimize pruning ratio allocation, we introduce a layer-adaptive sparsity assignment method that dynamically allocate the global pruning ratio based on empirical reconstruction gains across layers. Our empirical analysis uncovers intriguing insights into depth-wise and module-wise redundancy patterns, offering actionable insights for future LLM pruning designs. Through comprehensive experiments on diverse LLM benchmarks, we show that our proposed NeuroSlice method consistently surpasses state-of-the-art structured pruning baselines.

## 1 Introduction

Large language models (LLMs) have become foundational to natural language processing (NLP), driving unprecedented advancements across diverse applications (Vaswani et al., 2017; Brown et al., 2020; Grattafiori et al., 2024; Hui et al., 2024). However, the impressive performance of these models comes at the significant cost of computational and memory resources, restricting their widespread deployment (Achiam et al., 2023; Lin et al., 2024). To mitigate these substantial resource demands, pruning techniques represent one of the most prevalent methods for compressing LLMs, effectively reducing model size and computational demands while maintaining performance integrity (Ma et al., 2023; Sun et al., 2024). Among the various pruning strategies, structured pruning (An et al., 2024; Gao et al., 2024) stands out for its hardware-friendly properties, making it particularly attractive for practical implementations of LLMs.

Existing pruning methods for LLMs (Sun et al., 2024; Ling et al., 2024; Yin et al., 2024) typically rely on layer outputs as a primary signal for estimating parameter importance. However, in this work, we question whether this layer-output granularity is sufficiently fine-grained: can we decompose it further to yield a more informative signal for pruning?

We introduce a novel neuron-centric, additive decomposition of the layer output. This approach provides a view that is significantly more fine-grained than standard layer outputs. Specifically, leveraging matrix identity transformations, we reformulate the layer output as an exact sum of individual neuron contributions, spanning feed-forward network (FFN) layers and attention layers. Here, a "neuron" denotes the smallest vector unit in within FFN layers and attention layers, and is finer than head-level granularity. This decomposition transforms the layer output from its original shape of $\mathbb{R}^{s \times d}$ into a decomposed form: $\mathbb{R}^{s \times d \times n}$, where $s$ is the sequence length and $d$ is the hidden dimension, and $n$ is the number of **neurons**. The original output is perfectly recovered by summing

along this new neuron dimension. Here, each slice along the neuron axis isolates a single neuron's contribution to the layer output, thereby preserving fine-grained information about its individual contribution. Consequently, by leveraging this granular signal, it facilitates more precise importance selection of LLM parameters, enabling targeted retention of informative neurons and superior model performance preservation.

This neuron-centric decomposition of the layer output provides a clear and intuitive motivation for our pruning strategy. By isolating the contribution of each individual neuron, we can directly assess their functional importance in reconstructing the layer's output. Consequently, we reframe the complex task of structured pruning as a more tractable **neuron subset selection problem**: for each layer, our goal is to identify the subset of neurons that can most accurately reconstruct the original layer's output. However, simply selecting neurons with the highest individual contributions is suboptimal, as it ignores the significant correlations and redundancies between them. To address this, we employ a sophisticated greedy forward selection strategy that iteratively adds the neuron most correlated with the unexplained output residual, leading to a simple and effective structured pruning method.

Furthermore, our forward selection framework provides a effective way to address another critical challenge in LLM pruning: the heterogeneous sensitivity of different layers to parameter removal. Instead of imposing a uniform sparsity ratio, which ignores the fact that some layers are more critical than others, our method naturally enables a global, cross-layer comparison of neuron importance. The marginal information gain , calculated as each neuron is selected, can be served as a normalized and universally comparable metric of its contribution. This allows us to preserve the top-ranked neurons globally until the pruning budget is exhausted. Our unified approach thus dynamically assigns higher keep-ratios to critical layers, tackling both local (redundancy) and global (layer sensitivity) challenges within a single, efficient framework.

The contribution of this work is summarized as follows:

- **Decomposition of the layer output.** We introduce neuron contribution decomposition method that reframes the conventional layer output as an additive sum of unified neuron contributions across FFN layers and attention layers (at a granularity finer than heads). This provides more fine-grained signal for pruning, facilitating more precise importance selection of LLM parameters.

- **Forward Selection Framework for Structured Pruning** Building on the decomposition, we reframe structured pruning as a neuron subset selection problem that iteratively retains neurons maximizing marginal residual contributions. This novel framework effectively addresses both local redundancies and global layer sensitivities through forward selection and adaptive sparsity allocation.

- **State-of-the-art pruning results and insights.** Our NeuroSlice method achieves superior zero-shot accuracy and lower perplexity compared to baselines. Furthermore, our analysis uncovers depth-wise and module-wise redundancy patterns, offering actionable insights for future LLM architecture and pruning designs.

## 2   ASSESSING EACH NEURON'S CONTRIBUTION TO THE LAYER OUTPUT

In this section, we address the following research questions to motivate and validate our neuron contribution decomposition (NeuCoDe) approach:

- **Layer outputs decomposition**: Can we decompose layer outputs into finer-grained signals, providing richer information for parameter importance estimation compared to layer activations?
- **Unified pruning granularity**: Can we disassemble both FFN layers and attention heads into unified, vectorized components to enable a consistent, surgical pruning unit across modules?
- **Empirical validation of neuron contributions**: Do these decomposed neuron effectively distinguish between important and redundant neurons?

### 2.1   NEURON CONTRIBUTION DECOMPOSITION

To address these research questions, we decompose the output of each LLM layer into additive contributions from individual neurons using matrix identity transformations. Our method builds upon

insights from mechanistic interpretability (Elhage et al., 2021; Geva et al., 2021; Zhou et al., 2024), and we extend it further by novelly decomposing attention heads into finer-grained components, which we term LLM neurons.

To clearly define the concept, an "***LLM neuron***" is defined as a weight vector of length equal to the hidden dimension $d$, corresponding to a column (or row, depending on the matrix orientation) in the projection matrices of FFN and attention layers. In FFN layers, *LLM neurons* align with the intermediate channels, as defined in the literature (Wei et al., 2024). In attention layers, neurons represent finer-grained weight vectors within each head's query, key, value, and output matrices, enabling decomposition beyond head-level granularity. This unified definition facilitates precise importance assessment and a consistent processing pipeline for structured pruning.

**FFN Layer:** In line with prior research by Geva et al. (2021; 2022), we decompose the outputs of FFN layers explicitly as the sum of individual neuron outputs. Specifically, given an input $\mathbf{X}^\ell \in \mathbb{R}^{s \times d}$, where $s$ is the sequence length and $d$ is the hidden size, and the FFN up-projection, gating, and down-projection matrices $\mathbf{U}^\ell, \mathbf{G}^\ell, \mathbf{D}^\ell \in \mathbb{R}^{d_{\mathrm{FFN}} \times d}$, the FFN output can be derived as through matrix identity transformation:

$$
\begin{aligned}
\mathbf{Y}^\ell \quad &= \quad \mathrm{FFN}^\ell(\mathbf{X}^\ell) = \left( f(\mathbf{X}^\ell (\mathbf{U}^\ell)^\top) \ \odot \ \mathbf{X}^\ell (\mathbf{G}^\ell)^\top \right) \mathbf{D}^\ell \\
&= \quad \sum_{j=1}^{d_{\mathrm{FFN}}} \left[ f(\mathbf{X}^\ell \mathbf{u}_j^\ell) \odot \left( \mathbf{X}^\ell \mathbf{g}_j^\ell \right) \right] \mathbf{d}_j^\ell = \sum_{j=1}^{d_{\mathrm{FFN}}} \mathbf{m}_j^\ell \, \mathbf{d}_j^\ell
\end{aligned}
\tag{1}
$$

where $d_{\mathrm{FFN}}$ is the FFN intermediate dimension, $\mathbf{u}_j^\ell$, $\mathbf{g}_j^\ell$, and $\mathbf{d}_j^\ell$ denote the $j$-th columns of $\mathbf{U}^\ell$, $\mathbf{G}^\ell$, and $\mathbf{D}^\ell$, respectively, and $f(\cdot)$ is the activation function.

**Attention Layer:** Prior mechanistic interpretability studies have shown attention layer outputs can be computed as the sum of outputs of individual attention heads. However, attention heads remain relatively coarse-grained units, each encompassing substantial parameters at the *matrix level*. To achieve even finer-grained analysis, we further decompose each attention head into *neurons* using matrix identity transformations. Consequently, attention weights and attention outputs are expressed as sums of individual neuron contributions.

Specifically, for the $\ell$-th layer with input $\mathbf{X}^\ell \in \mathbb{R}^{s \times d}$, the attention layer is parameterized by four matrices $\mathbf{W}_Q^\ell, \mathbf{W}_K^\ell, \mathbf{W}_V^\ell, \mathbf{W}_O^\ell \in \mathbb{R}^{d \times d}$. These projection matrices are split into $H$ attention heads: $\mathbf{W}_Q^{\ell,i}, \mathbf{W}_K^{\ell,i}, \mathbf{W}_V^{\ell,i} \in \mathbb{R}^{d \times (d/H)}$ and $\mathbf{W}_O^{\ell,i} \in \mathbb{R}^{(d/H) \times d}$. The attention output is:

$$
\mathrm{Att}^\ell(\mathbf{X}^\ell) = \mathbf{Concat} \left[ \mathbf{A}^{\ell,1} \mathbf{X}^\ell \mathbf{W}_V^{\ell,1}, \ \mathbf{A}^{\ell,2} \mathbf{X}^\ell \mathbf{W}_V^{\ell,2}, \ \ldots, \ \mathbf{A}^{\ell,H} \mathbf{X}^\ell \mathbf{W}_V^{\ell,H} \right] \mathbf{W}_O^\ell = \sum_{i=1}^{H} \mathbf{A}^{\ell,i} (\mathbf{X}^\ell \mathbf{W}_V^{\ell,i}) \mathbf{W}_O^{\ell,i}
$$

where $\mathbf{A}^{\ell,i} = \mathrm{softmax} \left( \frac{(\mathbf{X}^\ell \mathbf{W}_Q^{\ell,i})(\mathbf{X}^\ell \mathbf{W}_K^{\ell,i})^\top}{\sqrt{d/H}} \right)$.

To Further decompose attention heads into finer-grained LLM neurons, we apply matrix identity transformations. Let $\mathbf{w}_Q^{\ell,i,j}, \mathbf{w}_K^{\ell,i,j}, \mathbf{w}_V^{\ell,i,j}, \mathbf{w}_O^{\ell,i,j}$ denote the $j$-th neuron in the respective matrices for head $i$. Then:

$$
\mathbf{A}^{\ell,i} \quad = \quad \mathrm{softmax} \left( \frac{(\mathbf{X}^\ell \mathbf{W}_Q^{\ell,i})(\mathbf{X}^\ell \mathbf{W}_K^{\ell,i})^\top}{\sqrt{d/H}} \right) = \mathrm{softmax} \left( \frac{\sum_{j \in \mathcal{J}_h} \mathbf{q}_j^{\ell,i} \, \mathbf{k}_j^{\ell,i^\top}}{\sqrt{d_h}} \right),
\tag{2}
$$

$$
\mathrm{Att}^\ell(\mathbf{X}^\ell) \quad = \quad \sum_{i=1}^{H} \mathbf{A}^{\ell,i} (\mathbf{X}^\ell \mathbf{W}_V^{\ell,i}) \mathbf{W}_O^{\ell,i} = \sum_{i=1}^{H} \mathbf{A}^{\ell,i} \sum_{j=1}^{d_h} \mathbf{v}_j^{\ell,i} \, \mathbf{o}_j^{\ell,i} = \sum_{i=1}^{H} \sum_{j=1}^{d_h} A_j^{\ell,i} \, \mathbf{v}_j^{\ell,i} \, \mathbf{o}_j^{\ell,i}
\tag{3}
$$

where $\quad \mathbf{q}_j^{\ell,i} \ = \ \mathbf{X}^\ell \mathbf{w}_Q^{\ell,i,j}, \qquad \mathbf{k}_j^{\ell,i} = \mathbf{X}^\ell \mathbf{w}_K^{\ell,i,j}, \qquad \mathbf{v}_j^{\ell,i} = \mathbf{X}^\ell \mathbf{w}_V^{\ell,i,j}, \qquad \mathbf{o}_j^{\ell,i} = \mathbf{w}_O^{\ell,i,j}$

To this end, we have decomposed the outputs of FFN layers, attention layers, and attention weights into sums of contributions from LLM neurons in the corresponding layers. This introduces a fine-grained information dimension to guide LLM parameter selection, which we term **neuron contribution decomposition (NeuCoDe)**. We compare this with conventionally adopted LLM layer activations below:

The aggregated layer activation collapses all latent neuron contributions into one, whereas the NCD matrix retains every neuron contribution separately, providing more fine-grained information compared to conventional LLM activations.

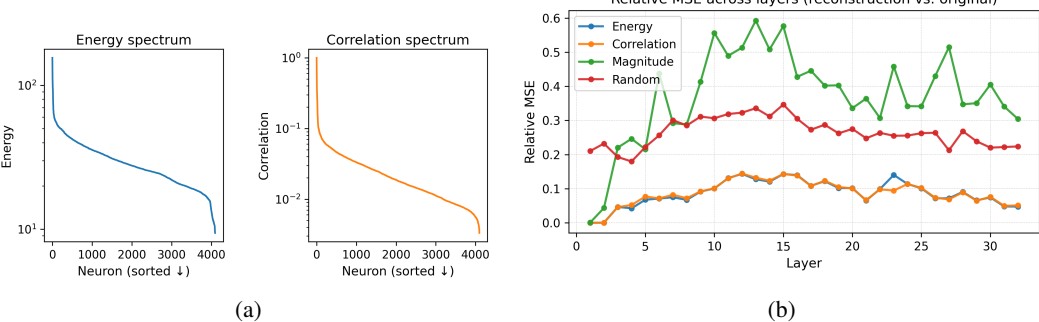

Figure 1: Neuron contribution scores distribution across neurons in an attention layer (a), and relative MSE of pruning based on four different criteria (b).

## 2.2 Do Neuron Contribution Distinguish Important from Redundant Neurons?

After obtaining per-neuron contributions via NeuCoDe, we investigate whether these contributions provide a reliable ranking signal to distinguish salient neurons from redundant ones. We conduct two prior experiments on LLaMA-2-7B, using 64 calibration samples and another 64 evaluation samples from WikiText-2 (1024 tokens each).

We examine neuron outputs across each layer of the LLM. To quantify inter-neuron differences, we adopt two criteria: *neuron energy* and *neuron correlation*. *Neuron energy* is defined as the average L2 norm of a neuron's output over all tokens in calibration samples. *Neuron correlation* is defined as the average absolute correlation between the neuron's output and the overall layer output over each token in calibration samples.

**Neuron Score distribution.** To assess the distribution of neuron importance within a layer, we compute both *neuron energy* and *neuron correlation* for each neuron. Figure 1a visualizes the per-neuron scores for an attention layer containing $4,096$ neurons, with values sorted in descending order. The plots reveal substantial variance across neuron scores, indicating that neurons do not contribute uniformly in the lens of neuron contribution. This suggests that the proposed criteria can effectively separate neurons in a layer.

**Accuracy of Selected Neurons.** While the above scores suggest a clear separation of neurons in LLM layers, we next ask: *do these scores translate into accurate selection of important LLM neurons*? Specifically, can they guide the pruning of neurons while maintaining model fidelity?
To validate this, we retain only the top $70\%$ of neurons in each attention layer based on different selection criteria: neuron energy, neuron correlation, neuron weight magnitude, and random selection. For each pruned variant, we compute the relative mean squared error (MSE) between the original layer output and the reconstructed output after pruning.
Figure 1b illustrate the experimental results. Both neuron energy and neuron correlation–based selections consistently outperform the magnitude-based and random baselines, yielding significantly lower relative MSE across all layers. This indicates that neuron contributions derived from NeuCoDe indeed serve as reliable signals for identifying and preserving the most informative neurons.

## 3 NeuroSlice: Forward Selection–Based Neuron Subset Selection

The previous section verified that neuron contribution decomposition provides a novel and informative signal for distinguishing critical neurons from redundant ones. Building on this, we now leverage the signal to perform structured pruning of LLMs. Concretely, we cast pruning as a **neuron subset selection** problem and solve it with a forward selection strategy inspired by classical feature selection. The resulting method is termed NeuroSlice.

### 3.1 PROBLEM FORMULATION

Let $\mathbf{X}^\ell \in \mathbb{R}^{s \times d}$ be the input tokens to the $\ell$-th Transformer layer and $\mathbf{Y}^\ell \in \mathbb{R}^{s \times d}$ its layer output. Applying the NeuCoDe in Section 2.1, we obtain a set of neuron outputs

$$\mathbf{N}^\ell = \left[\mathbf{N}_1^\ell, \ldots, \mathbf{N}_{n_\ell}^\ell\right], \quad \mathbf{N}_j^\ell \in \mathbb{R}^{N \times d}, \qquad \mathbf{Y}^\ell = \sum_{j=1}^{n_\ell} \mathbf{N}_j^\ell. \tag{4}$$

where $n^\ell$ is the number of neurons in layer $\ell$.

**Goal.** We seek a subset of LLM neurons that best reconstructs the original layer output. Let $\mathcal{S}\ell \subseteq \{1, \ldots, n^\ell\}$ be the index set of the selected neurons with $|\mathcal{S}\ell| = k$. The objective is

$$\min_{\mathcal{S}_\ell : |\mathcal{S}_\ell| = k} \left\| \mathbf{Y}^\ell - \sum_{j \in \mathcal{S}_\ell} \mathbf{N}_j^\ell \right\|_F^2. \tag{5}$$

We find equation 5 is essentially a cardinality-constrained feature subset selection task, where neuron outputs serve as "features" and $\mathbf{Y}^\ell$ is the regression target. We apply a forward selection strategy to identify neuron subsets for each layer.

### 3.2 FORWARD SELECTION OF INFORMATIVE NEURONS

**Core idea.** Forward selection is a greedy subset selection strategy: starting with an empty set, it iteratively adds the candidate that maximally reduces the current reconstruction error. In our setting, each candidate is a neuron output $\mathbf{N}_j^\ell$, and the goal is to approximate $\mathbf{Y}^\ell$ using only $k$ of them, as in equation 5.

**Greedy criterion.** Forward selection builds $\mathcal{S}_\ell$ one neuron at a time. At iteration $t$, the residual is

$$\mathbf{r}^{(t)} = \mathbf{Y}^\ell - \sum_{j \in \mathcal{S}_\ell^{(t)}} \mathbf{N}_j^\ell,$$

and the next neuron is the one whose output is most correlated with this residual:

$$j^\star = \arg\max_{j \notin \mathcal{S}_\ell^{(t)}} \left| \langle \mathbf{N}_j^\ell, \mathbf{r}^{(t)} \rangle \right|. \tag{6}$$

This process iteratively explains the largest remaining portion of $\mathbf{Y}^\ell$ until $|\mathcal{S}_\ell| = k$.

**Computational bottleneck.** A naive implementation requires nested loops: an outer loop over the $k$ iterations and an inner loop over the remaining neurons to recompute correlations with the residual each time. This leads to repeated traversals over the neuron outputs, making it computationally intensive for billion-parameter LLMs.

**Residual-update acceleration.** To address this, inspired by (Wang et al., 2012), we precompute the Gram matrix and initial correlations once:

$$\mathbf{G} = (\mathbf{N}^\ell)^\top \mathbf{N}^\ell \quad \text{and} \quad \mathbf{b} = (\mathbf{N}^\ell)^\top \mathbf{Y}^\ell,$$

At each step of the forward selection, the correlations with the residual are given by $\mathbf{b}$, since $\langle \mathbf{N}_j^\ell, \mathbf{r}^{(t)} \rangle = b_j$ (initially) and updates as follows: after selecting the $j^\star$ neuron, subtract the contribution of $\mathbf{N}_{j^\star}^\ell$ from all remaining correlations:

$$\mathbf{b} \leftarrow \mathbf{b} - \mathbf{G}_{:, j^\star}, \qquad \mathbf{b}_{j^\star} \leftarrow 0,$$

This reduces the nested computations to a one-time initialization followed by simple updates, eliminating the need of the inner loop iterations and greatly decreasing execution time. Empirically this scheme is 3–4 × faster than the naive forward selection of neurons. This streamlined forward-selection pipeline retains good efficiency, which is validated in ablations in Section 4.4.

**One pass for all sparsities.** Forward selection produces a complete ranked list of neurons by their marginal contributions. Once computed once per model, any sparsity level can be achieved by selecting the top-ranked neurons accordingly, without rerunning the procedure. This provides substantial efficiency gains over methods requiring repruning or fine-tuning for each sparsity target.

### 3.3 LAYER-WISE PRUNING RATIO ASSIGNMENT

Instead of imposing a uniform sparsity ratio, which ignores the fact that some layers are more critical than others (Yin et al., 2024). Our pruning method naturally enables a global, cross-layer comparison of neuron importance. Specifically, we allocate neurons *proportionally to their marginal information gain* obtained during forward selection, yielding layer-specific pruning ratios.

**Measuring marginal gain.** Forward selection already records the per-layer MSE after each kept neuron, $\{\text{MSE}_\ell^{(t)}\}$. To normalize gains, we define the information contributed by the $t$-th neuron in layer $\ell$ as

$$\Delta_\ell^{(t)} \;=\; \log\Big(\frac{\text{MSE}_\ell^{(t-1)}}{\text{MSE}_\ell^{(t)}}\Big). \tag{7}$$

These $\Delta_\ell^{(t)}$ values form an information profile indicating which additional neurons contribute most globally.

**Allocation rule.** We sort all candidate neurons across all LLM layers in *descending* order of $\Delta_\ell^{(t)}$ and keep them until the global budget $k_{\text{tot}}$ is met, while enforcing soft per-layer bounds to avoid extreme allocations. Let $\bar{k} = k_{\text{tot}}/L$ be the average keep count. We constrain each layer to $[\alpha\bar{k},\, \beta\bar{k}]$ ($\alpha = 0.8$, $\beta = 1.2$).

This procedure aligns the pruning ratio with the information richness of different blocks, preserving model performance from the perspective of neuron marginal contributions. No additional model passes are required, as per-neuron MSE values are obtained during forward selection.

## 4 EXPERIMENTS

We empirically evaluate NeuroSlice to address the following research questions:

- **Effectiveness.** How much accuracy / perplexity can pruned LLMs retain using NeuroSlice?
- **Pruning insights.** What structural patterns emerge from the forward–selection across model depth and between neuron types?
- **Efficiency.** What is the computational cost of our NeuroSlice method relative to existing methods?
- **Ablations.** How do different components of our method contribute to its overall effectiveness?

### 4.1 EXPERIMENTAL SETUP

**Calibration data.** Following prior work (Ashkboos et al., 2024; Frantar & Alistarh, 2023), we draw 128 sequences of length 2048 from the WIKITEXT-2 (Merity et al., 2016) training split as the calibration corpus. All baselines use the identical calibration set unless their original implementation mandates otherwise.

**Evaluation.** *(i) Zero-shot reasoning.* We report accuracy on five widely used commonsense and science reasoning benchmarks: PIQA (Bisk et al., 2020), WinoGrande (Sakaguchi et al., 2021), HellaSwag (Zellers et al., 2019), ARC-Easy, and ARC-Challenge (Clark et al., 2018), using the official `lm-eval-harness` framework (Biderman et al., 2024). *(ii) Language generation.* We evaluate the generation capabilities of the pruned models by perplexity (PPL) on the WIKITEXT-2 validation split.

**Models.** We evaluate three popular open-source LLMs, including: LLaMA-7B (Touvron et al., 2023a), LLaMA-2-7B (Touvron et al., 2023b), and LLaMA-3-8B (Grattafiori et al., 2024).

**Baselines.** In addition to the dense model, we compare our proposed **NeuroSlice** with state-of-the-art structured pruning baselines, including *SliceGPT* (Ashkboos et al., 2024), *FLAP* (An et al., 2024), *LoRAP* (Li et al., 2024), *LLM-BIP* (Wu, 2024), *Wanda-SP*, the structured version of (Sun et al., 2024), *SlimLLM* (Huang et al., 2025), and *CFSP* (Wang et al., 2025).

The baseline methods are not compatible with all the models we evaluated; therefore, we report

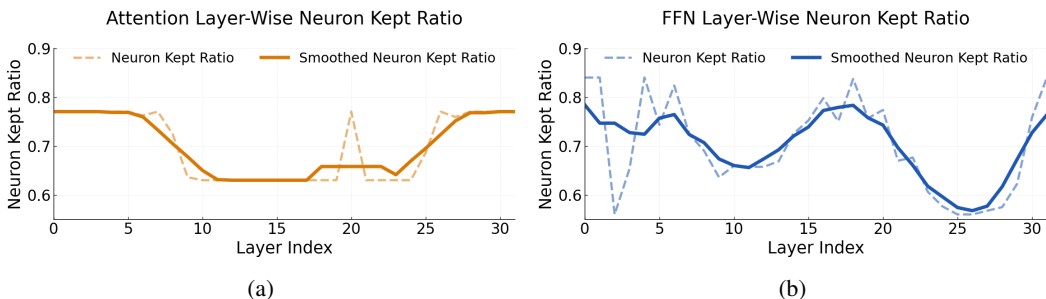

(a)                (b)

Figure 2: Layer-wise Neuron Kept Ratio for (a) Attention layers and (b) FFN layers after applying layer-wise pruning ratio assignment. Dashed curves show the raw layer-wise ratios; solid curves show the average values over a 5-layer window to highlight overall trends.

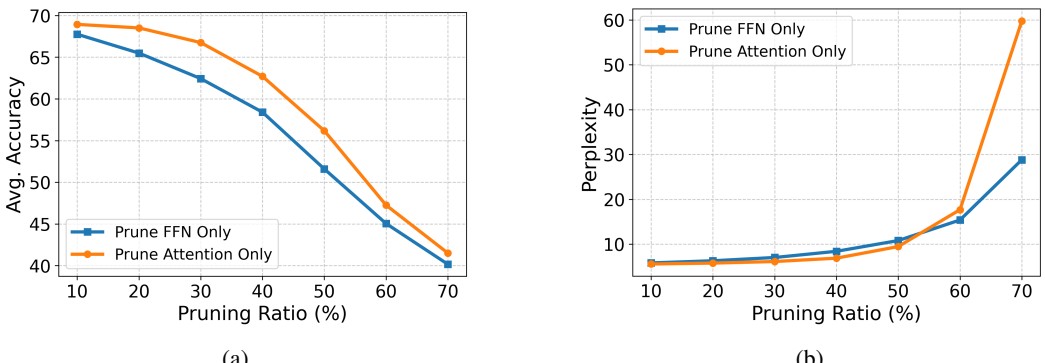

(a)                (b)

Figure 3: Comparison of pruning only FFN layers vs. only Attention layers at different overall pruning ratios. (a) Accuracy consistently degrades faster when pruning FFN layers only. (b) Perplexity grows more rapidly once pruning attention layers exceeds mid-level sparsity, indicating attention capacity becomes a sharper bottleneck at high pruning ratios.

their results only on the models they support. Pruning results of baselines are mainly based on the experimental results reported in (Wang et al., 2025; Huang et al., 2025; Wu, 2024).

Unless specified, sparsity refers to the percentage of parameters removed per layer. Following prior works (Wang et al., 2025; Huang et al., 2025), we report pruning results at $20\%$ and $50\%$ sparsity in the main experiment.

## 4.2 OVERALL RESULTS

Table 1 reports zero-shot accuracy on five reasoning benchmarks and language-modeling perplexity (PPL) at sparsity levels of $20\%$ and $50\%$ following prior works (Wang et al., 2025; Huang et al., 2025). At $20\%$ sparsity, **NeuroSlice** achieves the best average accuracy on all three models, retaining over **90%** of the dense accuracy in every case. At $50\%$ sparsity, NeuroSlice remains the best on LLaMA-7B and LLaMA-2-7B, and is competitive on LLaMA-3-8B. Overall, our NeuroSlice method yields gains of roughly 1-5 accuracy points over the strongest baseline in most settings.

In terms of perplexity, NeuroSlice exhibits very low perplexity at $20\%$ sparsity and maintains stable next-token predictive quality even at $50\%$ sparsity. Across all three models and both sparsity levels, it matches or outperforms competing methods. For example, NeuroSlice achieves less than half the perplexity of SlimLLM (Huang et al., 2025) on LLaMA-7B and LLaMA-2-7B, , indicating superior preservation of generative capability. Overall, these results indicate that selecting neurons via our contribution-based forward selection preserves both reasoning accuracy and language modeling quality more effectively than prior structured pruning baselines.

Table 1: Overall performance after pruning. ↓ = lower-is-better (perplexity), ↑ = higher-is-better (accuracy).

| Model | Sparsity | Method | PPL↓ | PIQA | Wino | Hella | ARC-e | ARC-c | Avg.↑ |
|---|---|---|---|---|---|---|---|---|---|
| | 0% | *Dense* | 5.67 | 79.16 | 69.93 | 76.2 | 72.90 | 44.80 | 68.60 |
| LLaMA-7B | 20% | FLAP | 7.27 | 76.28 | 66.61 | 68.39 | 68.06 | 40.36 | 63.94 |
| | 20% | LoRAP | 15.69 | 76.44 | 65.9 | 69.98 | 60.56 | 38.48 | 62.27 |
| | 20% | LLM-BIP | 16.98 | 77.85 | 66.92 | 71.29 | 64.98 | 39.84 | 64.17 |
| | 20% | SlimLLM | 15.95 | 75.95 | 66.06 | 69.82 | 64.48 | 39.33 | 63.13 |
| | 20% | **NeuroSlice** | **7.11** | 76.71 | 67.09 | 71.37 | 69.65 | 40.36 | **65.04** |
| | 50% | FLAP | 17.28 | 66.93 | 58.73 | 50.01 | 48.07 | 30.08 | 50.76 |
| | 50% | LoRAP | 56.96 | 63.82 | 57.3 | 46.96 | 40.36 | 27.73 | 48.85 |
| | 50% | LLM-BIP | 32.67 | 67.68 | 56.70 | 49.81 | 45.62 | 28.42 | 49.65 |
| | 50% | SlimLLM | 37.89 | 65.40 | 58.80 | 49.94 | 45.83 | 30.38 | 50.07 |
| | 50% | **NeuroSlice** | **13.31** | 65.98 | 59.93 | 53.39 | 51.01 | 31.76 | **52.42** |
| | 0% | *Dense* | 5.47 | 79.11 | 69.06 | 75.99 | 74.58 | 46.25 | 68.90 |
| LLaMA-2-7B | 20% | SliceGPT | **6.64** | 69.48 | 65.27 | 59.16 | 60.31 | 37.71 | 58.39 |
| | 20% | FLAP | 7.10 | 74.76 | 62.67 | 65.04 | 61.62 | 37.03 | 60.22 |
| | 20% | LoRAP | 15.02 | 76.39 | 65.11 | 69.15 | 61.99 | 35.58 | 61.64 |
| | 20% | SlimLLM | 15.70 | 76.28 | 63.54 | 68.88 | 65.74 | 39.08 | 62.70 |
| | 20% | **NeuroSlice** | 7.02 | 76.05 | 65.90 | 71.37 | 71.59 | 43.88 | **65.76** |
| | 50% | SliceGPT | **14.35** | 61.45 | 55.46 | 49.44 | 54.46 | 28.83 | 47.17 |
| | 50% | FLAP | 16.86 | 67.74 | 57.14 | 50.50 | 39.77 | 28.5 | 48.73 |
| | 50% | LoRAP | 60.89 | 62.23 | 55.41 | 43.98 | 38.51 | 27.65 | 45.56 |
| | 50% | SlimLLM | 38.64 | 64.74 | 53.28 | 45.91 | 39.73 | 29.01 | 46.53 |
| | 50% | **NeuroSlice** | 15.58 | 66.72 | 56.11 | 54.58 | 55.90 | 32.46 | **53.17** |
| | 0% | *Dense* | 6.13 | 80.74 | 72.45 | 79.16 | 77.74 | 53.41 | 72.70 |
| LLaMA-3-8B | 20% | SliceGPT | 10.89 | 63.33 | 63.38 | 52.17 | 51.81 | 33.28 | 52.79 |
| | 20% | Wanda-SP | 9.39 | 75.41 | 67.56 | 65.99 | 65.40 | 41.38 | 63.15 |
| | 20% | FLAP | 9.40 | 74.65 | 65.67 | 62.41 | 61.36 | 35.15 | 59.85 |
| | 20% | CFSP | 8.97 | 77.64 | 70.32 | 72.74 | 68.10 | 43.86 | 62.70 |
| | 20% | **NeuroSlice** | **8.08** | 76.99 | 70.72 | 70.42 | 67.42 | 43.17 | **65.74** |
| | 50% | SliceGPT | 45.94 | 49.19 | 50.24 | 28.50 | 31.29 | 18.72 | 35.59 |
| | 50% | Wanda-SP | 19.49 | 63.98 | 59.51 | 45.71 | 44.95 | 27.99 | 48.43 |
| | 50% | FLAP | 21.06 | 62.35 | 58.80 | 41.89 | 40.28 | 26.11 | 45.89 |
| | 50% | CFSP | 17.45 | 66.76 | 62.04 | 49.96 | 48.74 | 30.89 | **51.68** |
| | 50% | **NeuroSlice** | **17.21** | 66.37 | 61.35 | 49.99 | 48.09 | 30.45 | 51.35 |

## 4.3 NEURON-LEVEL PRUNING ANALYSIS

Building on the neuron-level pruning results, we seek more insights by probing two fine-grained questions here. **(i)** How is parameter redundancy distributed over layer depth? **(ii)** Which sub-modules—attention or FFN—offer greater pruning head-room before accuracy and perplexity deteriorate?

- **Depth-wise structure.** Figure 2 plots the post-pruning *neuron-kept ratio* for LLaMA-2-7B after applying our adaptive layer-wise allocation (its effectiveness is ablated in Section 4.4). FFN layers exhibit a pattern that early and mid-depth FFNs retain a larger share of important neurons, whereas later FFNs tolerate heavier pruning. Attention layers display a different U-shaped curve: both the first and the final attention layers contains more important nuerons. This aligns with prior intuition: early attention distributes lexical context broadly, while late attention sharpens task-specific resolution; mid-depth attention heads are more redundant.

- **Attention vs. FFN redundancy.** To decouple the two sub-modules, we prune *only* FFN layers or *only* attention layers at various global sparsity budgets. Ecxperimental results are depicted in Figure 3. From 10% to 50% sparsity, pruning attention layers alone incurs markedly smaller accuracy loss and perplexity growth than pruning FFNs alone, indicating higher removable redundancy in attention. Beyond 50% sparsity, however, the trend reverses: excessive pruning of attention heads causes a sharp perplexity spike, suggesting that attention capacity becomes the primary bottleneck once mid-level redundancy is exhausted.

### 4.4 ABLATION STUDY

We ablate key design choices of NeuroSlice to answer the following questions: **(Q1)** Does adaptive layer-wise sparsity assignment alleviate the performance degradation of pruned models? **(Q2)** Does the proposed neuron-level forward selection outperform a one-pass direct ranking? **(Q3)** What is the end-to-end pruning efficiency of NeuronSlice compared with the existing method? **(Q4)** How sensitive is NeuroSlice to the size of the calibration dataset?

**Adaptive Layer-Wise Sparsity Assignment.** We compare NeuroSlice with and without the adaptive layer-wise sparsity allocation at a fixed global pruning ratio. For the *non-adaptive* variant, each pruned layer keeps an identical per-layer retention ratio (= global retention). Perplexities for multiple LLMs are reported in Table 2 of Appendix B.1. Adaptive sparsity allocation consistently lowers perplexity of pruned models, demonstrating the effectiveness of our approach and highlighting the inherently uneven sparsity distribution across layers.

**Forward Selection vs. Direct Selection.** We contrast our iterative, correlation-aware forward selection with a one-pass *direct selection* baseline that removes neurons having the lowest individual contribution scores. Figure 4 of Appendix B.2 plots average zero-shot accuracy of LLaMA2-7B model across different pruning ratios. Forward selection yields higher accuracy at every sparsity. This gap reflects that direct ranking overestimates redundant but correlated neurons, whereas our forward selection method effectively mitigates this issue. Despite the iterative nature of forward selection, our implementation is efficiency-optimized: both strategies finish neuron selection in roughly ∼20 minutes under the same hardware setting. These findings support the necessity of our neuron forward selection method when targeting higher sparsity regimes.

**End-to-End Pruning Efficiency.** To demonstrate the efficiency of our method, we benchmark the end-to-end pruning time against the SliceGPT (Ashkboos et al., 2024) method, and present the experimental results in Table 3 of Appendix B.3. Despite using an accurate forward-selection procedure, NeuroSlice achieves pruning times comparable to SliceGPT. The efficiency comes from (i) eliminating any recovery fine-tuning and (ii) accelerating forward selection. A further advantage in efficiency is that NeuroSlice only needs one pass for all sparsities: NeuroSlice computes a single global ordering of neurons in one pass, after which any sparsity level can be achieved by selecting the top-ranked neurons accordingly, without rerunning the pruning computations. Consequently, the incremental cost of producing $K$ different sparsity variants is effectively $O(1)$ after the initial pass. In contrast, existing methods often require reconstruction and fine-tuning per target ratio.

**Calibration Dataset Size.** We evaluate the robustness of our method across different calibration size. As illustrated in Figure 5 of Appendix B.4, model perplexity improves sharply up to 128 tokens and then plateaus, suggesting diminishing returns once contribution estimates stabilize. Therefore, we apply a calibration size as 128 in experiments.

Additionally, details of the related work are provided in Appendix A.

## 5 CONCLUSION

We introduced **NeuroSlice**, a structured pruning framework that reframes layer-output based importance estimation by decomposing each layer's output into additive, unified neuron contributions across FFN and attention. Building upon this, we cast pruning as neuron subset selection solved via an accelerated forward-selection procedure with adaptive, cross-layer sparsity allocation. Empirically, NeuroSlice consistently preserves zero-shot reasoning accuracy and language-modeling quality better than strong baselines across three LLMs and different sparsities. Additionally, our method offers practical efficiency benefits: one pass yields a global neuron ordering that serves all sparsity targets without reruns. Our analysis further reveals depth-wise and module-wise redundancy patterns, providing actionable insights for future architecture and compression design.

## 6 REPRODUCIBILITY STATEMENT

We are committed to full reproducibility. Upon publication, we will release all code in this work. Further environment details will also be included in the public repository to ensure a smooth replication process.

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

## A  RELATED WORK

**LLM pruning.** To reduce the memory and latency of Transformer LLMs, represent one of the most prevalent methods for compressing LLM (Zhu et al., 2024). Unstructured pruning can yield greater compression ratios (Sun et al., 2024; Mason-Williams & Dahlqvist, 2024), but its irregular sparsity makes efficient hardware acceleration challenging. Structured pruning is preferred for hardware efficiency. It has progressed from gradient-guided layer or column removal (Ma et al., 2023) to row or column deletion with dense re-factoring (Ashkboos et al., 2024), fluctuation-based one-shot channel/column criteria (An et al., 2024), and sub-layer–aware compression that mixes low-rank in attention with structured pruning in FFN (Li et al., 2024). Newer designs refine layerwise decisions via OBS-style batched greedy selection (Ling et al., 2024) dimension-independent structural rules that relax width coupling (Gao et al., 2024), and block-wise importance propagation to curb error accumulation (Wu, 2024).

**Pruning signals.** Existing pruning methods primarily rely on several key information dimensions to assess weight importance, broadly categorized into gradient-based methods (Ma et al., 2023), regularization-based methods (Guo et al., 2023), activation-based methods (e.g., (Sun et al., 2024; Yin et al., 2024)), second-order information-based methods (Frantar & Alistarh, 2023; Ling et al., 2024), magnitude-based methods (Sun et al., 2024), and structure-filtering methods specific to LLM architectures (Men et al., 2024; Tao et al., 2023). NeuroSlice instead decomposes the layer outputs into additive, unified "neuron" contributions across FFN and attention (finer than heads), yielding a new, model-intrinsic information dimension for importance estimation. This decomposition-plus-selection view both sharpens local neuron choices and adapts sparsity to layer sensitivity.

Table 2: Perplexity of pruned models generated with NeuroSlice, with and without adaptive layer-wise sparsity (ALS.

| Method | Sparsity | LLaMA-7B | LLaMA2-7B | LLaMA3-8B |
|--------|----------|----------|-----------|-----------|
| NeuroSlice **w/** ALS | 70% | 1642.19 | 358.48 | 285.94 |
| NeuroSlice **w/o** ALS | 70% | 2488.43 | 411.04 | 376.79 |

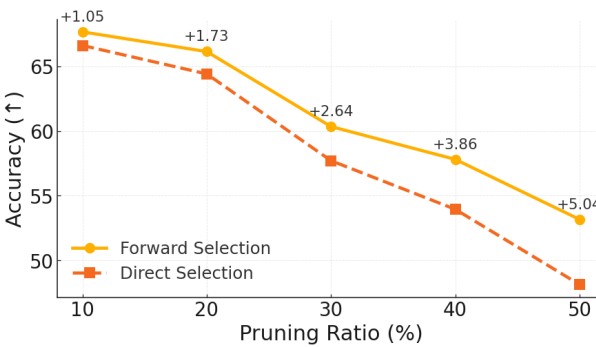

Figure 4: Ablation on neuron selection strategy. "*Forward Selection*" performs iterative inclusion based on incremental contribution, while "*Direct Selection*" removes neurons in a single pass by neuron contribution score. Forward selection yields consistently higher accuracy.

# B  ABLATIONS

## B.1  ADAPTIVE LAYER-WISE SPARSITY ASSIGNMENT

We compare NeuroSlice with and without the adaptive layer-wise sparsity allocation at a fixed global pruning ratio in Figure 2.

## B.2  FORWARD SELECTION VS. DIRECT SELECTION

We contrast our iterative, correlation-aware forward selection with a one-pass direct selection baseline that removes neurons having the lowest individual contribution scores in Figure 4.

## B.3  PRUNING-TIME COMPARISON

Table 3: Time required to prune LLAMA2-7B to 20% sparsity with identical hardware. NeuroSlice produces a global neuron ordering in a single pass; pruning for any desired sparsity are then cached without re-running pruning.

| Model | Method | One-time prune | Total time for $K$ sparsities |
|-------|--------|----------------|-------------------------------|
| LLAMA2-7B | NeuroSlice | 1h 15m | $\approx$ 1h 15m |
| | SliceGPT | 1h 7m | $\approx K \times$ 1h 12m |

To demonstrate the efficiency of our method, we benchmark the end-to-end pruning time against the SliceGPT (Ashkboos et al., 2024) method. Experiments conducted on one NVIDIA H100-80GB GPU. Our method show significant efficiency advantage in pruning $K$ sparsities for a model.

## B.4  CALIBRATION DATASET SIZE

We evaluate the robustness of our method across different calibration size in Figure 5.

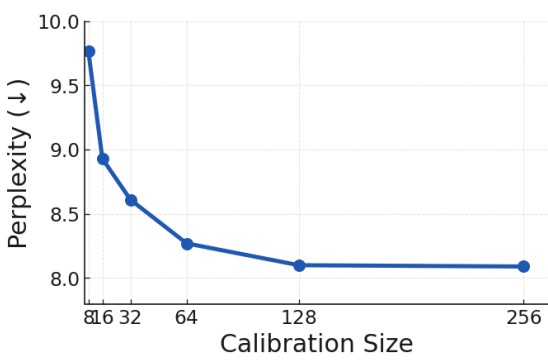

Figure 5: Impact of calibration-set size on validation perplexity.

## C   THE USE OF LLMs

We only use LLMs for polishing writing.

