# OpenReview forum: "NeuroSlice: Forward Selection-Based LLM Pruning via Neuron Contribution Decomposition"
_ICLR.cc/2026/Conference — Submitted to ICLR 2026_

### Official Review · Reviewer_6C4V · 2025-10-26

**Soundness:** 3
**Presentation:** 3
**Contribution:** 3
**Rating:** 4
**Confidence:** 4

**Summary:**

The paper proposes NeuroSlice, a structured pruning method for LLMs that first decomposes each layer’s output into additive per-neuron contributions across both FFN and attention modules, then casts pruning as a neuron subset selection problem solved by forward selection. A ranked list of neurons is produced by iteratively adding the neuron most correlated with the current residual, with an efficient residual-update scheme based on a precomputed Gram matrix to avoid repeated passes. The method also allocates layer-wise pruning ratios by distributing neurons in proportion to their marginal information gains recorded during selection.

**Strengths:**

1. The paper introduces a decomposition that reframes layer outputs as sums of per-neuron contributions for both FFN and attention at finer granularity than heads, yielding a consistent pruning unit and a richer importance signal.

2. Marginal information gains recorded during selection drive cross-layer pruning ratios, acknowledging that some layers are more critical than others.

3. On multiple LLaMA variants, NeuroSlice matches or outperforms baselines on zero-shot accuracy while achieving substantially lower perplexity, especially at 20 percent and 50 percent sparsity.

**Weaknesses:**

1. The paper studies structured pruning but does not report end-to-end acceleration on GPUs at different sparsity levels. It would be important to show latency, throughput, and energy measurements on real hardware across a range of sparsity targets.
2. The method’s sensitivity to the choice of calibration dataset is underexplored. A systematic analysis across domains, sequence lengths, and calibration budgets would clarify how robust the neuron ranking is to distribution shift.
3. The novelty may be limited because the importance signal is derived from reconstruction of activations. The approach reads as a finer-grained activation-based criterion rather than a fundamentally new importance measure, so the conceptual advance should be articulated more clearly.

**Questions:**

1. The current evaluation only covers the LLaMA family. Please include additional model families such as Qwen and Mixtral to demonstrate robustness across architectures and training recipes.

2. The strategy for determining layer-wise sparsity lacks comparison to OWL (outlier-based) and similar methods, as well as deeper insights about where sparsity should be allocated. Additional analysis could reveal layer trends and help explain when and why certain layers are pruned more aggressively.

3. The approach stores a score for every neuron. Please report the memory overhead of these scores, describe how it scales with model size and sequence length, and provide experiments on larger models (for example, beyond 30B parameters) to validate scalability in practice.

---

> ### Author Response · Authors · 2025-11-28
> **Author Response (1/2)**
>
> Dear Reviewer 6C4V,
>
> We thank the reviewer for the thoughtful feedback. We are encouraged by your recognition of our paper, including (i) the **a consistent pruning unit and a richer importance signal**, (ii) the insight that **certain layers are more critical than others**, and (iii) the **strength of our experimental evidence**. Below, we carefully address each of your **Weaknesses (W)**, **Questions (Q)** with corresponding **Answers (A)**. We will incorporate these clarifications into the revised paper to make the contributions and empirical evidence clearer.
>
> ---
> **[W1]:  Does not report end-to-end acceleration**
>
> **[A2]: Measurement of throughput and memory**
>
> Thank you for the constructive suggestion. We report end-to-end inference measurements on LLaMA-2-7B to demonstrate that NeuroSlice’s structured pruning translates into real acceleration. We measure **throughput** (tokens/s) and **memory** during decoding. Results are reported as *relative changes* vs. the dense model:
>
> | Sparsity | Throughput (tokens/s) | Memory |
> |---|---|---|
> | 20% | **1.32×**|**-19.68%**|
> | 50% | **1.69×**|**-49.32%**|
>
> ---
> **[W2]: Sensitivity to calibration data is underexplored**
>
> **[A2]: Ablation on calibration data**
>
> Thank you for the valuable feedback. We report ablations on calibration samples in **Sec. 4.4** and **Appx. B.4** and will expand them for clarity and breadth.
>
> Following most pruning works [1-3], the sequence length of each sample is 2048 in our work. As illustrated in Figure 5 of the paper, the model perplexity improves sharply up to using 128 calibration samples and then plateaus. This **sample efficiency** is noticeably better than FLAP [3] and SliceGPT [1], which require 1024 samples calibration.
>
> Following your suggestion, we will conduct ablations on the calibration dataset including WikiText2, Alpaca, and C4.
>
> **Reference**:
>
> [1] SliceGPT: Compress Large Language Models by Deleting Rows and Columns, ICLR 2024
>
> [2] SliM-LLM: Salience-Driven Mixed-Precision Quantization for Large Language Models, ICML 2025
>
> [3] Fluctuation-based Adaptive Structured Pruning for Large Language Models, AAAI 2024
>
> ---
> **[W3]: The conceptual advance should be articulated more clearly.**
>
> **[A3]: Core novelty: providing a strictly more informative pruning signal, thereby increasing the upper bound of pruning.**
>
> Thank you for the thoughtful feedback. We will explain this from both an *intuitive* and a *statistical* perspective.
>
> - **More informative signal.**
> Existing pruning methods largely use **layer outputs**, which aggregate all neuron contributions into a single tensor of shape *(token × feature)* per layer. In contrast, we explicitly decompose the layer into a tensor of **neuron contributions** with shape *(token × feature × neuron)*. The standard layer output can be obtained by summing along the neuron axis, so it is a **many-to-one transformation** of our statistic. This means that our pruning signal is **strictly more informative** in the sense of *statistical decision theory*: the standard layer outputs can be obtained by applying a measurable transformation to our neuron contribution statistic, but not vice versa.
>
> - **Intuitive view.**
>   Pruning is a selection problem: we must decide which atomic units to keep based on the information contained in the pruning signal. Different methods may exploit a given signal more or less effectively, however, **the upper bound of what any method can achieve is limited by the information the signal itself provides**. Because our decomposition provides strictly more information, it enlarges the space of “good” selection rules and raises the ceiling on what pruning methods can achieve.
>
> - **Statistical decision view.**
>   More formally, pruning can be cast as a **statistical decision problem**: given a pruning signal, a decision rule selects a subset of atomic units under loss against latent “true importance”. The best achievable performance given a pruning signal is its **minimum achievable risk**. Our neuron contribution signal is an **exact additive decomposition** of the layer output, while the layer-output signal is a deterministic post-processing of it. By **Blackwell’s notion of informativeness**, the neuron-contribution statistic *dominates* the layer-output statistic. Therefore, its minimum achievable risk is **strictly smaller**. Equivalently, **no** method that only sees layer outputs can surpass the performance upper bound achievable when using the decomposed neuron contributions.
>
>   Under finite calibration data, this increased informativeness manifests as **tighter Fano/Chernoff-type error bounds** (lower mis-selection probability) and **higher Fisher information** (lower variance) for importance estimates. Practically, this reduces rank inversions among neurons and stabilizes the selection process.
>
> In light of your feedback, we will clarify these points in the revised version to better distinguish our framework from prior pruning methods

---

> ### Author Response · Authors · 2025-11-28
> **Author Response (2/2)**
>
> **[Q1]: Please include additional model families.**
>
> **[A4]: Additional experiments on larger and non-LLaMA models (OPT-13B&OPT-30B).**
>
> Following your insightful suggestion, we have extended our experiments from the LLaMA family to the OPT series, including OPT-13B and OPT-30B, further demonstrating the effectiveness of NeuroSlice.
>
> Table 2: NeuroSlice vs. SliceGPT on OPT models (20%/50% sparsity).
>
> |Model|Sparsity.|Method|PPL↓|PIQA|Wino|Hella|ARC-e|ARC-c|Avg.↑|
> |---|---|---|---|---|---|---|---|---|---|
> |OPT-13B|0%|Dense|10.12|76.82|64.80|69.81|61.87|35.67|61.79|
> |OPT-13B|20%|SliceGPT|10.75|74.48|64.96|65.42|60.90|35.24|60.20|
> |OPT-13B|20%|NeuroSlice|10.50|76.03|65.04|66.51|61.45|35.07|60.82
> |OPT-13B|50%|SliceGPT|15.39|64.41|56.83|45.90|44.44|27.82|47.98|
> |OPT-13B|50%|NeuroSlice|15.18|71.78|59.80|53.19|52.66|31.14|**53.71 (+5.73)**|
> |OPT-30B|0%|Dense|9.56|78.07|68.19|72.27|65.24|38.23|64.40|
> |OPT-30B|20%|SliceGPT|9.91|76.50|66.61|70.61|64.18|35.75|62.73|
> |OPT-30B|20%|NeuroSlice|9.88|77.97|67.40|70.46|64.91|37.64|63.67|
> |OPT-30B|50%|SliceGPT|12.47|68.16|59.76|52.28|50.25|30.01|52.09|
> |OPT-30B|50%|NeuroSlice|11.96|71.47|62.70|61.38|56.03|32.85|**56.89 (+4.80)**|
>
> ---
> **[Q2]: Strategy determining layer-wise sparsity lacks comparison to OWL (outlier-based) and similar methods; additional analysis could reveal layer trends**
>
> **[A5]:**
> **Comparison to OWL/outlier-based allocation.** :
> Our allocation is driven by **marginal information gain** at each selection step, which makes cross-layer contributions comparable on a common scale. We agree with reviewer that making a comparison with the outlier-based methods is helpful. We will add a direct comparison to outlier-based allocation under equal global sparsity.
>
> **Additional Analysis**: We analyzed the layer sparsity trends in Sec. 4.3 of the paper and reveal two findings:
>
> - **FFN layers.** Early and mid‑depth FFNs have a larger share of important neurons, while later FFNs tolerate heavier pruning.
>   *Interpretation:* A widely held view is that FFN layers store and inject factual knowledge into the representation. In early and middle layers, FFNs layers add knowledge-bearing features into the input activations that downstream attention can route. In later layers, the network shifts toward higher-level reasoning and decision consolidation rather than further knowledge injection.
>
> - **Attention layers.** We observe a U-shaped pattern: the early and final attention layers concentrate more important neurons.
>   *Interpretation:* Early attention performs lexical/context routing (critical for downstream layers), while late attention make high-level reasoning and decision consolidation; mid-depth attention is more redundant and thus pruned more.
>
> - **Sparsity regime effect.** Below 50% sparsity, attention-only pruning degrades less than FFN-only pruning, suggesting greater redundancy in attention. Beyond 50%, the trend reverse.
>
>   *Interpretation:* Attention’s **routing** role tolerates moderate pruning better than the FFN’s **knowledge-injection** role. However, at high sparsity, damaging the routing role has disproportionately severe consequences compared to further trimming FFN units.
>
> ---
> **[Q3]:** The approach stores a score for every neuron. Please report the memory overhead of these scores.
>
> **[A6] Memory overhead of these scores is small (LLaMA-2-7B example).**
>
> **Neuron Counts**:
> - Layer number: 32
> - FFN neurons per layer: 11008 * 3 (up/gate/down projections)
> - Attention neurons per layer: 4096 * 4 (q/k/v/o projections)
> - All neurons**: 1,581,056
>
> Given that each neuron score is stored in float 32, which is 4 bytes, the overall memory overhead of these scores is ~**6 MiB** for LLaMA-2-7B, which can be easily scaled to different model sizes.
>
> Additionally, we have validated our method on **OPT-30B in [A4]**, showcasing the scalability in practice.
>
> ---
> We are encouraged to find that these experiments and enhancements, prompted by your valuable insights, have substantially strengthened our paper. We look forward to hearing your feedback!

---

### Official Review · Reviewer_esWz · 2025-10-31

**Soundness:** 4
**Presentation:** 4
**Contribution:** 3
**Rating:** 8
**Confidence:** 4

**Summary:**

This paper introduces a neuron pruning method called NeuroSlice which involves neuron contribution decomposition (NeuCoDe), neuron subset selection, and flexible one shot pruning. Two importance metrics are introduced, neuron energy and neuron correlation. These metrics demonstrate improved selection criteria over baselines of magnitude selection and random selection. Neuron subset selection is performed via forward selection, which involves an efficient iterative selection of neurons that greedily minimizes the reconstruction error. This algorithm ranks all neurons and therefore can be used to achieve any sparsity target without additional computation.This method outperfoms other structured pruning approaches at comparable efficiency. Ablation studies and analysis validate the findings and highlight interesting trends.

**Strengths:**

The paper is very well written. The method development is easy to follow and experiments validate the main points. The method is simple, interesting, and effective. The results are strong and demonstrate the efficacy of the method.

**Weaknesses:**

Some minor typos. See questions.

**Questions:**

1. What are the comparable pruning times for other methods?
2. Have neuron energy and neuron correlation been used in other pruning methods? Or are the authors the first to use these metrics?
3. How does the method perform on non-Llama models? I understand if these results cannot be obtained during the rebuttal period but it would be interesting to see results on a different model family.

---

> ### Author Response · Authors · 2025-11-28
> **Author Response**
>
> Dear Reviewer esWz,
>
> We thank the reviewer for the thoughtful feedback. We are encouraged by your recognition of our paper, including  (i) **the method is simple, interesting, and effective**,  (ii) **the results are strong**, (iii)**the method development is easy to follow**, and (iv) **the paper is very well written**.
>
> Below, we carefully address each of your **Questions (Q)** with corresponding **Answers (A)**. We will incorporate these clarifications into the revised paper to make the contributions and empirical evidence clearer.
>
> ---
> **[Q1]: What are the comparable pruning times for other methods?**
>
> **[A1]: Runtime comparison**
>
> Thank you for the valuable question. We reported the end-to-end runtime of NeuroSlice in Sec. 4.4 and Appendix B.3 of the paper. For your convenience, we have also included it here. NeuroSlice constructs **one global ordering** in **a single pass**, then **caches** it. Hence **one-time cost** is comparable to SliceGPT, but **total pruning time for K target sparsities** improves substantially:
>
> Table 1: Runtime comparison
>
> | Method        | One-time prune (single sparsity) | Total time for \(K\) sparsities |
> |:---|:----|:---|
> | **NeuroSlice**|1h15min| ≈ **1×** one-time-pruning (independent of K) |
> | SliceGPT|1h7min| ≈ **K ×** one-time-pruning (re-prune per sparsity) |
>
> ---
> **[Q2]: Have “neuron energy” and “neuron correlation” been used in other pruning methods?**
>
> **[A2]** Thank you for the question. To the best of our knowledge, **Neuron Contribution Decomposition** is the **first** work in pruning that (i) **decomposes** the layer output into additive per-neuron contributions and (ii) applies the same neuron formulation to both FFN and attention, yielding a **consistent pruning unit across submodules** (finer than heads).
>
> - **Why this matters.** Since the conventional layer output signal is a deterministic post-processing (sum) of our statistic. In the sense of *statistical decision theory* (Blackwell informativeness), neuron contribution is **strictly more informative** than layer outputs. Consequently, the minimum achievable pruning risk under neuron contributions is lower than that under layer outputs. Equivalently, using neuron contribution signal **raises the upper bound** of pruning methods that use the signal of layer output.
>
> - **On “energy/correlation.”** We are uncertain if *energy* and *correlation* have appeared in other contexts. However, our work is, to our knowledge, the **first to define and use “neuron energy” and “neuron correlation” on the *decomposed neuron-contribution signal***.
>
> We will make this distinction explicit in the revision.
>
> ---
> **[Q3]: How does the method perform on non-LLaMA models?**
>
> **[A3]: Additional experiments on larger and non-LLaMA models (OPT-13B&OPT-30B).**
>
> Following your insightful suggestion, we have extended our experiments from the LLaMA family to the OPT series, including OPT-13B and OPT-30B, further demonstrating the effectiveness of NeuroSlice.
>
> Table 2: NeuroSlice vs. SliceGPT on OPT models (20%/50% sparsity).
>
> |Model|Sparsity.|Method|PPL↓|PIQA|Wino|Hella|ARC-e|ARC-c|Avg.↑|
> |---|---|---|---|---|---|---|---|---|---|
> |OPT-13B|0%|Dense|10.12|76.82|64.80|69.81|61.87|35.67|61.79|
> |OPT-13B|20%|SliceGPT|10.75|74.48|64.96|65.42|60.90|35.24|60.20|
> |OPT-13B|20%|NeuroSlice|10.50|76.03|65.04|66.51|61.45|35.07|60.82
> |OPT-13B|50%|SliceGPT|15.39|64.41|56.83|45.90|44.44|27.82|47.98|
> |OPT-13B|50%|NeuroSlice|15.18|71.78|59.80|53.19|52.66|31.14|**53.71 (+5.73)**|
> |OPT-30B|0%|Dense|9.56|78.07|68.19|72.27|65.24|38.23|64.40|
> |OPT-30B|20%|SliceGPT|9.91|76.50|66.61|70.61|64.18|35.75|62.73|
> |OPT-30B|20%|NeuroSlice|9.88|77.97|67.40|70.46|64.91|37.64|63.67|
> |OPT-30B|50%|SliceGPT|12.47|68.16|59.76|52.28|50.25|30.01|52.09|
> |OPT-30B|50%|NeuroSlice|11.96|71.47|62.70|61.38|56.03|32.85|**56.89 (+4.80)**|
>
> ---
> We are encouraged to find that these experiments and enhancements, prompted by your valuable insights, have substantially strengthened our paper. We look forward to hearing your feedback!

---

### Official Review · Reviewer_3bwS · 2025-11-01

**Soundness:** 2
**Presentation:** 2
**Contribution:** 2
**Rating:** 4
**Confidence:** 3

**Summary:**

NeuroSlice is a structured pruning method that treats pruning as neuron subset selection. It introduces Neuron Contribution Decomposition to measure each neuron’s additive impact on layer outputs, then applies forward selection to choose neurons that best preserve reconstruction error and determine adaptive sparsity per layer. The approach achieves comparable or slightly better perplexity and zero-shot performance on LLaMA-7B/2-7B/3-8B models than structured pruning baselines like SliceGPT, FLAP, SlimLLM, and CFSP.

**Strengths:**

1. The neuron-level decomposition is clearly formulated, linking pruning to feature selection with a mathematically explicit view of neuron contribution.

2. The same contribution formulation applies to both FFN and attention modules, offering a consistent pruning unit across submodules.

**Weaknesses:**

1. While the paper presents a clean formulation, the main idea—ranking neurons via contribution correlation and selecting them greedily—is a direct adaptation of feature-selection and orthogonal matching pursuit methods. Similar per-channel or per-head selection ideas appear in earlier works like SliceGPT, DISP-LLM, and even SparseGPT’s layerwise reconstruction; NeuroSlice mostly adds finer-grained bookkeeping and a new interpretation layer.

2. The results report perplexity and zero-shot accuracy but omit any measurement of latency, throughput, memory, or FLOPs. Without showing real-world acceleration (e.g., on vLLM, TensorRT-LLM, or FlashAttention kernels), it is unclear whether the structured pruning translates into practical efficiency.

3. Forward selection, even with Gram-matrix caching, still requires multiple matrix–vector correlations per layer. The paper claims it’s “3–4× faster” than naive selection, but provides no absolute runtime or GPU budget. For multi-billion-parameter LLMs, this cost could still be prohibitive relative to one-shot pruning or magnitude baselines.

**Questions:**

1. How does the computational cost (wall-clock time, GPU hours) of NeuroSlice compare to SliceGPT or FLAP on the same models?

2. Do you observe real inference acceleration (tokens/sec, latency) after pruning on actual inference frameworks (e.g., vLLM)?

3. I would suggest adding more open-source models (now only LlaMA family) to enhance the empirical studies. Some great candidates include Qwen3, Gemma, GPT OSS, etc.

---

> ### Author Response · Authors · 2025-11-28
> **Author Response (1/2)**
>
> Dear Reviewer 3bwS,
>
> We thank reviewer for the thoughtful feedback. We appreciate your recognition that (i) our **neuron-level decomposition is clearly formulated**, and (ii) the **same neuron-contribution formulation applies to both FFN and attention**, providing a **consistent pruning unit across submodules**. Below, we carefully address each of your **Weaknesses (W)** and **Questions (Q)** with corresponding **Answers (A)**. We will incorporate these clarifications into the revised paper to make the contributions and empirical evidence clearer.
>
> ---
> **[W1]: Contribution of NeuroSlice**
>
> **[A1]: Our core novelty: providing a strictly more informative pruning signal, thereby increasing the upper bound of pruning.**
>
> Thank you for the thoughtful feedback. Essentially, many pruning methods share a common *procedure*: ranking atomic units of LLMs using a pruning signal and select them greedily. The key sources of improvement, therefore, are *(i) how to effectively use the given pruning signals* and *(ii) whether one can provide a new and more informative signal*. NeuroSlice belongs to (ii): we **provide a more informative pruning signal**. We explain this from both an *intuitive* and a *statistical* perspective.
>
> - **More informative signal.**
> Existing pruning methods largely use **layer outputs**, which aggregate all neuron contributions into a single tensor of shape *(token × feature)* per layer. In contrast, we explicitly decompose the layer into a tensor of **neuron contributions** with shape *(token × feature × neuron)*. The standard layer output can be obtained by summing along the neuron axis, so it is a **many-to-one transformation** of our statistic. This means that our pruning signal is **strictly more informative** in the sense of *statistical decision theory*: the standard layer outputs can be obtained by applying a measurable transformation to our neuron contribution statistic, but not vice versa.
>
> - **Intuitive view.**
>   Pruning is a selection problem: we must decide which atomic units to keep based on the information contained in the pruning signal. Different methods may exploit a given signal more or less effectively, however, **the upper bound of what any method can achieve is limited by the information the signal itself provides**. Because our decomposition provides strictly more information, it enlarges the space of “good” selection rules and raises the ceiling on what pruning methods can achieve.
>
> - **Statistical decision view.**
>   More formally, pruning can be cast as a **statistical decision problem**: given a pruning signal, a decision rule selects a subset of atomic units under loss against latent “true importance”. The best achievable performance given a pruning signal is its **minimum achievable risk**. Our neuron contribution signal is an **exact additive decomposition** of the layer output, while the layer-output signal is a deterministic post-processing of it. By **Blackwell’s notion of informativeness**, the neuron-contribution statistic *dominates* the layer-output statistic. Therefore, its minimum achievable risk is **strictly smaller**. Equivalently, **no** method that only sees layer outputs can surpass the performance upper bound achievable when using the decomposed neuron contributions.
>
>   Under finite calibration data, this increased informativeness manifests as **tighter Fano/Chernoff-type error bounds** (lower mis-selection probability) and **higher Fisher information** (lower variance) for importance estimates. Practically, this reduces rank inversions among neurons and stabilizes the selection process.
>
> In light of your feedback, we will clarify these points in the revised version to better distinguish our framework from prior pruning methods
>
> ---
> **[W2&Q2]: Omit the measurement of latency**
>
> **[A2]: Measurement of throughput and memory**
>
> Thank you for the constructive suggestion. We report end-to-end inference measurements on LLaMA-2-7B to demonstrate that NeuroSlice’s structured pruning translates into real acceleration. We measure **throughput** (tokens/s) and **memory** during decoding. Results are reported as *relative changes* vs. the dense model:
>
> Table 1: Measurement of latency
>
> | Sparsity | Throughput (tokens/s) | Memory |
> |---|---|---|
> | 20% | **1.32×**|**-19.68%**|
> | 50% | **1.69×**|**-49.32%**|

---

> ### Author Response · Authors · 2025-11-28
> **Author Response (2/2)**
>
> ---
> **[W3&Q1]: absolute runtime of the method**
>
> **[A3]: Runtime comparison (here and Appendix B.3 of the paper)**
>
> Thank you for the valuable question. We reported the end-to-end runtime of NeuroSlice in Sec. 4.4 and Appendix B.3 of the paper. For your convenience, we have also included it here. NeuroSlice constructs **one global ordering** in **a single pass**, then **caches** it. Hence **one-time cost** is comparable to SliceGPT on the same GPU, but **total pruning time for K target sparsities** differs substantially:
>
> Table 2: Runtime comparison
>
> | Method        | One-time prune (single sparsity) | Total time for \(K\) sparsities |
> |:---|:----|:---|
> | **NeuroSlice**|1h15min| ≈ **1×** one-time-pruning (independent of K) |
> | SliceGPT|1h7min| ≈ **K ×** one-time-pruning (re-prune per sparsity) |
>
> ---
> **[Q3]: Adding more open-source models to enhance the empirical studies**
>
> **[A4]: Additional experiments on larger and non-LLaMA models (OPT-13B&OPT-30B).**
>
> Evaluation on more open-source models can indeed strengthen the empirical claims of our paper. Following your suggestion, we have extended our experiments from the LLaMA family to the OPT series, including OPT-13B and OPT-30B
>
> Table 3: NeuroSlice vs. SliceGPT on OPT models (20%/50% sparsity).
>
> |Model|Sparsity.|Method|PPL↓|PIQA|Wino|Hella|ARC-e|ARC-c|Avg.↑|
> |---|---|---|---|---|---|---|---|---|---|
> |OPT-13B|0%|Dense|10.12|76.82|64.80|69.81|61.87|35.67|61.79|
> |OPT-13B|20%|SliceGPT|10.75|74.48|64.96|65.42|60.90|35.24|60.20|
> |OPT-13B|20%|NeuroSlice|10.50|76.03|65.04|66.51|61.45|35.07|60.82
> |OPT-13B|50%|SliceGPT|15.39|64.41|56.83|45.90|44.44|27.82|47.98|
> |OPT-13B|50%|NeuroSlice|15.18|71.78|59.80|53.19|52.66|31.14|**53.71 (+5.73)**|
> |OPT-30B|0%|Dense|9.56|78.07|68.19|72.27|65.24|38.23|64.40|
> |OPT-30B|20%|SliceGPT|9.91|76.50|66.61|70.61|64.18|35.75|62.73|
> |OPT-30B|20%|NeuroSlice|9.88|77.97|67.40|70.46|64.91|37.64|63.67|
> |OPT-30B|50%|SliceGPT|12.47|68.16|59.76|52.28|50.25|30.01|52.09|
> |OPT-30B|50%|NeuroSlice|11.96|71.47|62.70|61.38|56.03|32.85|**56.89 (+4.80)**|
>
> ---
> We are encouraged to find that these experiments and enhancements, prompted by your valuable insights, have substantially strengthened our paper. We look forward to hearing your feedback!

---

### Official Review · Reviewer_bYbc · 2025-11-03

**Soundness:** 2
**Presentation:** 1
**Contribution:** 2
**Rating:** 2
**Confidence:** 4

**Summary:**

The paper proposes a neuron-level structured pruning framework for large language models. It reformulates layer outputs into per-neuron contributions and applies a forward selection strategy for pruning. Experiments on LLaMA models show modest but consistent improvements over existing baselines.

**Strengths:**

1. The paper addresses a relevant and timely topic in LLM efficiency, focusing on structured pruning with an intuitive neuron-level formulation. The proposed method is simple and implementation-friendly, requiring no fine-tuning.

2. Empirical results across multiple reasoning benchmarks demonstrate that the approach maintains stable accuracy under moderate sparsity.

**Weaknesses:**

1. The experiments are conducted only on the LLaMA family (7B–8B models), which are relatively small by current LLM standards. The lack of evaluation on larger or more diverse architectures limits the generality and credibility of the empirical claims.

2. The idea of neuron-level decomposition has been discussed in prior pruning works such as WANDA and related studies on structured sparsity. The proposed “neuron contribution decomposition” appears to be a reformulation rather than a fundamentally new perspective or mechanism.

3. The paper provides no illustrative figure or schematic to clarify the proposed framework. Without visual explanation or pseudocode, it is difficult for readers to follow the main algorithmic pipeline and intuition. Also, it is not standard to put results into appendix and discuss them in the main paper for the entire section 4.4.

4. Although the method shows slightly better accuracy and perplexity than some baselines, the gains are modest and may fall within expected experimental variance. The improvements do not convincingly demonstrate a clear advance over existing structured pruning techniques.

Part of review is revised with LLM assistance.

**Questions:**

Please see weaknesses.

---

> ### Author Response · Authors · 2025-11-28
> **Author Response (1/2)**
>
> Dear reviewer bYbc,
>
> Thank you for your effort in reviewing the paper. We appreciate you find our method is **simple and implementation-friendly**, and **empirical results show stable accuracy under moderate sparsity**. Below is a summary of our answers (**A**) to the weaknesses (**W**) you raised.
>
> ---
>  **[W1]**: lack of evaluation on larger or more diverse architectures
> **A1: Additional experiments on larger and non-LLaMA models (OPT-13B&OPT-30B)**
>
> Evaluation on larger and more diverse architectures can strengthen the empirical claims of our paper. Following your suggestion, we have extended our experiments from the LLaMA family to the OPT series, including OPT-13B and OPT-30B
>
> Table 1: NeuroSlice vs. SliceGPT on OPT models (20%/50% sparsity).
>
> |Model|Sparsity.|Method|PPL↓|PIQA|Wino|Hella|ARC-e|ARC-c|Avg.↑|
> |---|---|---|---|---|---|---|---|---|---|
> |OPT-13B|0%|Dense|10.12|76.82|64.80|69.81|61.87|35.67|61.79|
> |OPT-13B|20%|SliceGPT|10.75|74.48|64.96|65.42|60.90|35.24|60.20|
> |OPT-13B|20%|NeuroSlice|10.50|76.03|65.04|66.51|61.45|35.07|60.82
> |OPT-13B|50%|SliceGPT|15.39|64.41|56.83|45.90|44.44|27.82|47.98|
> |OPT-13B|50%|NeuroSlice|15.18|71.78|59.80|53.19|52.66|31.14|**53.71 (+5.73)**|
> |OPT-30B|0%|Dense|9.56|78.07|68.19|72.27|65.24|38.23|64.40|
> |OPT-30B|20%|SliceGPT|9.91|76.50|66.61|70.61|64.18|35.75|62.73|
> |OPT-30B|20%|NeuroSlice|9.88|77.97|67.40|70.46|64.91|37.64|63.67|
> |OPT-30B|50%|SliceGPT|12.47|68.16|59.76|52.28|50.25|30.01|52.09|
> |OPT-30B|50%|NeuroSlice|11.96|71.47|62.70|61.38|56.03|32.85|**56.89 (+4.80)**|
>
> ---
> **[W2]**: neuron-level decomposition has been discussed in prior pruning works such as WANDA
>
> **[A2]: Complete difference in both mathematical formulation and fundamental contribution**
>
> Thank you for your feedback. We respectfully clarify that neuron contribution decomposition is fundamentally different from WANDA or related studies.
>
> **(1) Different mathematical formulation.**
> To the best of our knowledge, the mathematical derivations in **Sec. 2.1** of our paper have not appeared in WANDA or closely related works. A direct comparison between **Sec. 2.1** of our paper and **Sec. 3** of WANDA clearly shows that the underlying equations, assumptions, and derivation pipelines are completely different.
>
> **(2) Our core novelty: providing a strictly more informative pruning signal, thereby increasing the upper bound of pruning.**
>
> We explain this from both an *intuitive* and a *statistical* perspective.
>
> - **More informative signal.**
> Existing pruning methods largely use **layer outputs**, which aggregate all neuron contributions into a single tensor of shape *(token × feature)* per layer. In contrast, we explicitly decompose the layer into a tensor of **neuron contributions** with shape *(token × feature × neuron)*. The standard layer output can be obtained by summing along the neuron axis, so it is a **many-to-one transformation** of our statistic. This means that our pruning signal is **strictly more informative** in the sense of *statistical decision theory*: the standard layer outputs can be obtained by applying a measurable transformation to our neuron contribution statistic, but not vice versa.
>
> - **Intuitive view.**
>   Pruning is a selection problem: we must decide which atomic units to keep based on the information contained in the pruning signal. Different methods may exploit a given signal more or less effectively, however, **the upper bound of what any method can achieve is limited by the information the signal itself provides**. Because our decomposition provides strictly more information, it enlarges the space of “good” selection rules and raises the ceiling on what pruning methods can achieve.
>
> - **Statistical decision view.**
>   More formally, pruning can be cast as a **statistical decision problem**: given a pruning signal, a decision rule selects a subset of atomic units under loss against latent “true importance”. The best achievable performance given a pruning signal is its **minimum achievable risk**. Our neuron contribution signal is an **exact additive decomposition** of the layer output, while the layer-output signal is a deterministic post-processing of it. By **Blackwell’s notion of informativeness**, the neuron-contribution statistic *dominates* the layer-output statistic. Therefore, its minimum achievable risk is **strictly smaller**. Equivalently, **no** method that only sees layer outputs can surpass the performance upper bound achievable when using the decomposed neuron contributions.
>
>   Under finite calibration data, this increased informativeness manifests as **tighter Fano/Chernoff-type error bounds** (lower mis-selection probability) and **higher Fisher information** (lower variance) for importance estimates. Practically, this reduces rank inversions among neurons and stabilizes the selection process.
>
> In light of your feedback, we will clarify these points in the revised version to better distinguish our framework from prior activation-based pruning methods

---

> ### Author Response · Authors · 2025-11-28
> **Author Response (2/2)**
>
> ---
> **[W3]** Lack of pseudocode/schematic illustration of the method and some results are put in appendix.
>
> **[A3]:** Improving method clarity with pseudocode, and put results in main paper.
>
> Following your suggestion, we will add a concise pseudocode block summarizing the full NeuroSlice pipeline (neuron contribution decomposition, Gram matrix construction, accelerated forward selection, global allocation of neurons). This will be placed in Section 3 to make the algorithm easy to implement.
>
> We will move the main ablation plots/tables into the main paper, keeping only secondary details in the appendix. We believe these changes will improve readability of the paper.
>
> ---
> **[W4]:** The improvements do not convincingly demonstrate a clear advance over existing structured pruning techniques.
>
> **[A4]:** We respectfully clarify that the improvement of NeuroSlice is significant. Across all evaluated model and sparsity settings, *NeuroSlice* exceeds the strongest structured pruning baseline in average accuracy in **5 of 6** cases with margins of **+0.87, +1.66, +3.06, +4.44, +2.59**. The only exception is Llama-3-8B at 50% sparsity with -0.33. In parallel, NeuroSlice consistently achieves **lower perplexity** (PPL). For example, **less than half of LLM-BIP, LoRAP, and SlimLLM's PPL**.
>
> Considering that at 20% sparsity the dense-sparse gap is naturally small, yet NeuroSlice still achieves clear gains over competitive baselines. We also appreciate that **reviewer esWz** finds *the experimental results strong and evidencing efficacy*, and **reviewer 6C4V** notes that NeuroSlice *outperforms baselines on zero-shot accuracy while achieving substantially lower PPL*. Together these results indicate a meaningful advance over existing structured pruning approaches.
>
> ---
> We are encouraged to find that these experiments and clarifications, prompted by your insights, have strengthened our paper. We look forward to hearing your feedback!

---

### Author Response · Authors · 2025-12-04
**Rebuttal Summary**

We sincerely thank the AC and reviewers for the time and effort dedicated to evaluating our paper.

---
We are deeply grateful for the recognition of our work across reviews:

- **Novelty & Methodology.** Reviewers recognized that our method is **simple, interesting, and effective** (esWz), that it provides **a consistent pruning unit across both FFN and attention** (3bwS), and that the method provides **a consistent pruning unit and a richer importance signal** (6C4V).
- **Experimental Validation**: Experimental **results are strong and demonstrate the efficacy of the method** (esWz, 6C4V), the method is **simple, implementation-friendly** and **the approach maintains stable accuracy across multiple benchmarks**(bYbc).
- **Presentation.** The paper is **well written** and the development **easy to follow** (esWz), and the **neuron-level decomposition is clearly formulated** (3bwS).

---
We further summarize the experiments and clarifications in the rebuttal:

**1) Clarifying contribution.**  We highlight that our method provides a strictly more informative pruning signal, thereby increasing the upper bound of pruning. We support this from both an intuitive and a statistical perspective.

**2) Broader model coverage.**  Following suggestions, we report pruning results on **non-LLaMA** models (e.g., **OPT-13B/30B**), where NeuroSlice maintains lower PPL and higher zero-shot average accuracy.

**3) Practical acceleration**  We add end-to-end measurements on LLaMA-2-7B to demonstrate practical acceleration and memory saving in standard inference.

**4) Absolute pruning runtime** As detailed in Appendix B.3, the one-time runtime of NeuroSlice is comparable to SliceGPT, but the total runtime for pruning K target sparsities is much shorter.

---
Finally, we would like to further highlight the contributions of our work.

- **A more informative pruning signal** We decompose the layer output into additive per-neuron contributions consistently across FFN and attention. This signal is strictly more informative than the standard layer-output signal, thereby increasing the attainable upper bound of pruning methods.

- **Forward selection with adaptive sparsity allocation**: We recast structured pruning as neuron subset selection, iteratively retaining neurons with maximal marginal residual contribution. This addresses within-layer redundancy and allocates sparsity globally across layers.

- **State-of-the-art pruning results** NeuroSlice attains higher zero-shot accuracy and lower perplexity on **5 LLMs** across different sizes and sparsity regimes.

- **Insights into LLM structure.**  Our analyses reveal **depth-wise** and **module-wise** redundancy patterns, offering actionable insights for future architecture and pruning designs.

---
In light of valuable feedbacks, we find our work has been significantly enhanced. Again, we would like to thanks for the dedication throughout the review process.

---

### Meta-Review · Area_Chair_5xtR · 2026-01-07

**Summary:**

The suggested decision is primarily informed by three recurring concerns raised across reviews:
1. Limited Novelty. While reviewers (bYbc, 3bwS, 6C4V) acknowledged that the paper is technically sound and clearly formulated, several noted that the core methodology closely follows existing activation or reconstruction pruning paradigms. The use of neuron-level decomposition and greedy forward selection was often viewed as an incremental refinement of prior approaches, rather than a fundamentally new pruning principle. As a result, the conceptual advance was not considered sufficiently strong or clearly distinguished from existing work.
2. Insufficient model and architecture coverage. The experimental evaluation in the original submission focused primarily on relatively LLaMA models. Reviewers (bYbc, 3bwS, esWz, 6C4V) expressed concern that this narrow scope limits the generality of the empirical claims, particularly given the diversity of modern LLM architectures and training recipes.
3. Lack of convincing acceleration evidence. Although the paper targets efficiency through structured pruning, several reviewers (3bwS, 6C4V) pointed out that the evaluation emphasized perplexity and zero-shot accuracy, while providing limited or no evidence of real-world efficiency gains.

**Reviewer Concerns:**

The authors made a meaningful effort to address the lack of convincing acceleration evidence. By providing additional measurements on throughput and memory usage, they strengthened the empirical case that the proposed structured pruning can lead to practical efficiency improvements, at least in a controlled inference setting.

However, several issues remain only partially resolved.
First, the novelty concern persists. Although the authors clearly argue that the contribution lies in elevating the pruning signal to a more informative, neuron-level representation, whether this constitutes a sufficiently strong conceptual advance over prior activation-based pruning remains debatable.
Second, while additional experiments on OPT models improve coverage, the evaluation is still limited to a small number of closely related LLM families, leaving questions about robustness across a broader range of architectures and training paradigms.
Third, despite reporting throughput and memory gains, the paper still lacks comparisons on widely used real-world inference frameworks (e.g., vLLM, FlashAttention), which limits confidence in the claimed system-level acceleration benefits in practical deployment scenarios.
These remaining concerns continue to influence the overall assessment of the paper.

**Reviewer Scores:**

It is difficult to know with certainty how individual reviewers would have adjusted their scores after full discussion, but based on the rebuttal and review comments, the following assessment seems reasonable.

The rebuttal addressed a subset of the reviewers’ concerns, particularly those related to missing efficiency and acceleration evidence. Reviewers whose primary concerns focused on the lack of convincing efficiency validation would likely have viewed the additional throughput and memory results positively.

However, reviewers who raised concerns about the level of novelty or the limited coverage of model families would likely have maintained a similar stance. While the rebuttal clarified the authors’ perspective on novelty and added experiments on OPT models, these points remain partially unresolved and may not fully address the underlying concerns.

Overall, the rebuttal likely reduced uncertainty for some reviewers but did not fundamentally change the balance of opinions, particularly regarding conceptual novelty and breadth of validation.

---

### Decision · Program_Chairs · 2026-01-26

Reject